# *Enterococcus faecium* L-15 Cell-Free Extract Improves the Chondrogenic Differentiation of Human Dental Pulp Stem Cells

**DOI:** 10.3390/ijms20030624

**Published:** 2019-01-31

**Authors:** Hyewon Kim, Sangkyu Park, Kichul Kim, Seockmo Ku, Jeongmin Seo, Sangho Roh

**Affiliations:** 1Cellular Reprogramming and Embryo Biotechnology Laboratory, Dental Research Institute, BK21, Seoul National University School of Dentistry, Seoul 08826, Korea; khw0063@naver.com (H.K.); good0039@hanmail.net (S.P.); fpdh0839@naver.com (K.K.); 2Biomedical Research Institute, NeoRegen Biotech Co., Ltd., Gyeonggi-do 16614, Korea; 3Fermentation Science Program, School of Agriculture, College of Basic and Applied Sciences, Middle Tennessee State University, Murfreesboro, TN 37132, USA; seockmo.ku@mtsu.edu

**Keywords:** dental pulp stem cells (DPSCs), lactic acid bacteria, *Enterococcus faecium*, chondrogenic differentiation

## Abstract

Hyaline cartilage is a tissue of very low regenerative capacity because of its histology and limited nutrient supply. Cell-based therapies have been spotlighted in the regeneration of damaged cartilage. Dental pulp stem cells (DPSCs) are multipotent and are easily accessible for therapeutic purposes. In human gastrointestinal tracts, *Enterococcus faecium* is a naturally occurring commensal species of lactic acid bacteria. In this work, the human DPSCs were differentiated into chondrocytes using a chondrogenic differentiation medium with or without L-15 extract. We observed that chondrogenic differentiation improved in an *E. faecium* L-15 extract (L-15)-treated DPSC group via evaluation of chondrogenic-marker mRNA expression levels. In particular, we found that L-15 treatment promoted early-stage DPSC differentiation. Cells treated with L-15 were inhibited at later stages and were less likely to transform into hypertrophic chondrocytes. In L-15-treated groups, the total amount of cartilage extracellular matrix increased during the differentiation process. These results suggest that L-15 promotes chondrogenic differentiation, and that L-15 may be used for cartilage repair or cartilage health supplements. To our knowledge, this is the first report demonstrating the beneficial effect of L-15 treatment on chondrogenic differentiation.

## 1. Introduction

Hyaline cartilage is composed of chondrocytes and extracellular matrix, including collagen, proteoglycans, and hyaluronic acid. The tissue is commonly damaged by aging, trauma, inflammation, and degenerative disease [1]. Because of its avascular and aneural character (which makes it difficult to regenerate when damaged), cell-based therapy is an optimal treatment. This is seen when autologous chondrocytes are implanted in a damaged area [1,2]. In clinical cases, autologous chondrocyte implantation (ACI) is unlikely to cause immune rejection and is effective in treating cartilage defects [3,4]. However, the number of chondrocytes in the body is low, and there are limited chondrocytes available for use [5]. Due to the substantial limitations of ACI, mesenchymal stem cells (MSCs) are getting spotlight as a cell source for cartilage repair [1,6,7].

Stem cells are suitable for tissue regeneration because they can self-renew and differentiate into various types of cells [6]. Dental pulp stem cells (DPSCs) are neural crest-derived mesenchymal stem cells that can be obtained from the pulp tissue of the tooth [8]. Unlike isolating bone marrow-derived mesenchymal stem cells, DPSCs can be obtained by non-invasive methods because the stem cells are present in extracted teeth [9]. As multipotent stem cells, DPSCs are capable of differentiating into adipogenic, osteogenic, chondrogenic, and myogenic cells, and they have high proliferation rates [10]. Moreover, DPSCs have the potential for use in cell-based therapies for type 1 diabetes, neurological diseases, and immunodeficiency diseases [11,12,13].

Probiotics comprise a variety of microbial species found in the mammalian gastrointestinal tract. Lactic acid bacteria (LAB) are mainly used as probiotic strains. They have positive effects on human health, including protection of the host from infection through immunomodulatory effects on homeostasis, and include Lactobacilli, Streptococci, Bifidobacteria, and Enterococci species [14]. Among these LAB varieties, *Enterococcus faecium* has been shown to improve intestinal health and reduce serum cholesterol levels [15,16]. It is a mass-produced microorganism for commercial application in nutraceutical and food supplement markets. Various research groups have conducted studies on the effects of LAB on adipogenic and osteogenic differentiation [17,18]. Recently, it was reported that *E. faecium* has antioxidant and anti-inflammatory effects, both in vitro and in vivo [19,20]. *E. faecium* has not been reported to have side effects, so it may be useful for the treatment and prevention of cartilage defects. Moreover, to our knowledge, there are no reports on the effects of LAB on cartilage differentiation. In this study, we investigated the impact of *E. faecium* L-15 extract on chondrogenic differentiation.

## 2. Results

### 2.1. Human DPSC Characterization and Isolation

Although various cell types were initially observed, homogeneous populations of fibroblast-like cells were observed after passage 3 (Figure 1a,b). To investigate the properties of human DPSCs (hDPSCs), cells were analyzed by fluorescence-activated cell sorting (FACS). Dental pulp tissues were obtained from two different donors and FACS analysis was conducted with each sample. At passage 4, the hDPSCs expressed high levels of MSC markers (i.e., CD10 (92.48%), CD29 (100%), CD44 (100%), CD73 (100%), CD90 (100%) and CD105 (88.13%)), but low levels of hematopoietic and endothelial stem cell markers (i.e., CD14 (20.11%), CD31 (0.53%), CD34 (1.24%), and CD45 (0.82%)) (Figure 1c,d and Appendix A). At passage 8, the hDPSCs showed similar surface marker expression to that at passage 4 (Appendix A). Therefore, passage 4–8 cells were used for chondrogenic differentiation.

### 2.2. Effect of E. faecium L-15 Extract (L-15) on hDPSC Viability

The effect of L-15 extract on cell viability was assessed by the Water-soluble tetrazolium salt (WST) assay. L-15 extract was prepared at 10, 25, 50, 100, 200, and 300 μg/mL. As shown in Figure 2, hDPSC viability was significantly decreased by treatments of 100 μg/mL or more. This suggested that an L-15 extract concentration of 50 μg/mL was safe, and this concentration was used for subsequent experiments.

### 2.3. L-15 Extract Promotes Early-Stage Chondrogenic Differentiation

The hDPSCs were differentiated into chondrocytes using chondrogenic differentiation medium with or without L-15 extract. Total mRNA was extracted from the control group (L-15 extract-free) and the L-15 extract-treated group (LET) at days 3, 5, 7, 10, and 14 to observe gene expression changes (Figure 3). Using quantitative real-time PCR, we examined the expression of early-stage chondrogenic markers (i.e., *SRY* (sex-determining region Y), box 9 (*SOX9*), aggrecan (*ACAN*), and collagen type 2 alpha 1 (*COL2A1*)) and late-stage chondrogenic markers (i.e., collagen type 10 alpha 1 (*COL10A1*), runt-related transcription factor 2 (*RUNX2*), and matrix metallopeptidase 13 (*MMP13*)). The expression of *SOX9* increased until day 10, then decreased at day 14 in the control group. The expression of *COL2A1* and *ACAN* increased until day 14 in the control group. Expression levels of *SOX9*, *COL2A1*, and *ACAN* were significantly higher in the LET group than the control group for 14 days. Specifically, *SOX9* and *COL2A1* showed significantly higher expression than the control group at days 5, 7, 10, and 14. *ACAN* showed significantly higher expression than the control group at days 5, 7, and 10. *SOX9*, *ACAN*, and *COL2A1* exhibited the highest differences between the two groups on day 5. The control group expression of *COL10A1*, *RUNX2*, and *MMP13* increased during the differentiation process. In the LET group, *COL10A1*, *RUNX2*, and *MMP13* showed significantly higher expression than the control group on day 5. *MMP13* (a late-stage chondrogenic marker and also an extracellular matrix degradation marker) was elevated in the LET group versus the control group at day 5, but significantly decreased in the LET group by day 7.

### 2.4. L-15 Extract Promotes ECM Formation during Chondrogenic Differentiation

The expression level of GAG, a marker specific to the extracellular matrix of chondrocytes, was quantitatively analyzed. GAG accumulation in the control and LET groups tended to increase over 10 days. GAG accumulation was significantly and consistently higher in the LET group than the control group, but the difference between the two groups decreased over time (Figure 4). Alcian blue staining was performed to assess accumulation of proteoglycan (e.g., GAG and hyaluronic acid) levels in chondrogenic-differentiated cells. Stained samples were quantified by spectrophotometry (Figure 5a). Similar to the results with GAG assay, the alcian blue staining intensity increased steadily in both groups over 14 days and was significantly higher in the LET group versus the control group (Figure 5b).

## 3. Discussion

Hyaline cartilage defects do not directly impact human survival, but they severely affect quality of life. As the elderly population is increasing in many countries, the number of patients with cartilage defects will continue to rise [1]. However, the current treatments for cartilage defects are limited. To provide symptomatic relief, new fundamental therapies are needed [21]. Stem cell-based therapies for cartilage regeneration have been widely studied in recent years [5,22,23]. In this study, we showed that early-stage chondrogenic markers (*SOX9*, *COL2A1*, and *ACAN*) and late-stage chondrogenic markers (*COL10A1*, *RUNX2*, and *MMP13*) are expressed during chondrogenic differentiation of human DPSCs. When hDPSCs were cultured in chondrogenic differentiation medium with L-15 extract, the expression of early-stage chondrogenic markers tended to be higher in the LET group than in the control group. The initial expression of the late-stage chondrogenic markers *COL10A1* and *RUNX2* was significantly higher in the LET group than in the control group. However, this increased expression gradually decreased to a level similar to the control group.

Hypertrophy is induced in the late stages of in vitro chondrogenic differentiation and is one of the problems that still has to be overcome in cell-based therapy for cartilage repair [24]. Once a chondrocyte enters the hypertrophic state, it is difficult for it to return to normal chondrocyte because hypertrophic chondrocytes secrete MMPs and do not maintain chondrocyte homeostasis [25]. MMP13, a late-stage chondrogenic marker, is the main enzyme responsible for cartilage degradation [26]. Compared to other MMP family members, MMP13 is specific to the extracellular matrix degradation of connective tissue, cartilage collagen, and proteoglycans [26]. A variety of clinical studies have reported that patients with cartilage destruction have high MMP13 expression [27,28]. Many research groups have also found that MMP13-overexpressing transgenic mice display a cartilage-destruction phenotype, which suggests that increased MMP13 is associated with cartilage degradation [29,30]. Interestingly, the expression of *MMP13* was increased by treatment of L-15 extract during the early stages. On the other hand, the expression of *MMP13* in the LET group decreased significantly during the later stages. Although the expression of late-stage chondrogenic markers was upregulated during the early stages, it was not hypertrophic state since the early-stage chondrogenic markers maintained higher expression levels during the period of differentiation, and the expression of *MMP13* had significantly lower levels during the later stages in the LET group than in the control group [31]. Cartilage ECM provides structural and biochemical support to chondrocytes and consists of collagen, aggrecan, and networks of proteoglycans which are composed of GAGs and hyaluronic acid [32]. Recently, several studies have reported that the efficiency of chondrogenic differentiation is enhanced by using collagen membranes or a hyaluronic acid and cartilage ECM [33,34]. In this study, we found that GAG increased during the later stages of chondrogenic differentiation. The LET group accumulated more GAG than the control group. The quantification of proteoglycans (e.g., GAG and hyaluronic acid) using alcian blue staining similarly verified chondrocyte differentiation. In summary, this study examined the effect of an L-15 extract on chondrogenic cell differentiation. We found that the L-15 extract initially promoted chondrogenic differentiation and subsequently inhibited the expression of the extracellular matrix degradation enzyme. The L-15 extract promoted chondrogenic differentiation by increasing certain extracellular matrix components (e.g., GAG). *E. faecium* is a type of LAB present in the human gastrointestinal tract and commonly used to produce fermented meat or dairy products [35,36]. Several groups have used LAB to induce adipogenic or osteogenic differentiation [37,38]. However, this is the first study to apply LAB to induce chondrogenic differentiation and demonstrate the beneficial effect of *E. faecium* L-15 in chondrogenic differentiation.

## 4. Materials and Methods

### 4.1. Preparation of the E. faecium Extract

The *E. faecium* L-15 strain (KCTC13498BP) was used for this study and was obtained from NeoRegen Biotech (Suwon, Gyeonggi-do, Korea). This strain was originally isolated from a traditional Korean rice-fermented food containing flatfish. The L-15 strain was cultured in tryptic soy broth (TSB; Hardy Diagnostics, Santa Maria, CA, USA) and incubated for 18 h at 35 °C. The cultured L-15 strain was harvested, washed three times in phosphate-buffered saline (PBS), and resuspended in double-deionized water (ddH_2_O). The washed *E. faecium* was sonicated (Sonics, Stratford, CT, USA) on ice for 30 min. To remove the cellular debris, it was centrifuged at 12,000× *g* for 10 min. The supernatant was passed through a 0.45 μm filter and frozen at −80 °C overnight. It was then freeze-dried and reconstituted with PBS before use.

### 4.2. Isolation and Expansion of Human Dental Pulp Stem Cells (hDPSCs)

The study was conducted in accordance with the Declaration of Helsinki, and the protocol has been approved by the Institutional Review Board (IRB, number S-D20180004, 30 March 2018) at Seoul National University School of Dentistry. Informed consent was obtained from parents of all subjects prior to sample collection. Human maxillary central supernumerary teeth (*n* = 2) were extracted from children at the Dental Hospital of Seoul National University in accordance with the guidelines provided by the IRB. The hDPSC primary culture process followed our laboratory protocol. The extracted teeth were briefly cut around the cemento–enamel junction using a cutting disk. The pulp tissue was exposed and gently separated from the crown. The pulp tissue was minced into 1 mm^2^ pieces with a scalpel blade and transferred into 12-well culture dishes. The cells were then grown in Minimum essential medium eagle – α modification (α-MEM; Corning, Rochester, NY, USA) supplemented with 10% fetal bovine serum (FBS; PAN-Biotech, Bayern, Germany) and incubated in a 37 °C incubator with 5% CO_2_. The culture medium was replaced every three days. The cells from the different donors were cultured separately.

### 4.3. Characterization of hDPSCs by Fluorescence-Activated Cell Sorting (FACS)

At passage 4, the hDPSCs were detached and resuspended in ice-cold PBS containing 5% FBS. The cells were incubated on ice for 30 min with monoclonal antibodies against CD10-fluorescein isothiocyanate (FITC), CD29-Alexa 488, CD44-FITC, CD73-FITC, CD90-FITC, CD105-FITC, CD14-allophycocyanin (APC), CD34-Alexa 647, CD45-APC, and CD31-APC. All antibodies were purchased from Biolegend (San Diego, CA, USA). Analyses were performed using a FACSVerse (Becton Dickinson, Franklin Lakes, NJ, USA).

### 4.4. Chondrogenic Differentiation of hDPSCs

A pellet culture system was used for chondrogenic differentiation. The hDPSCs were trypsinized and resuspended in chondrogenic medium consisting of high-glucose DMEM supplemented with 50 μg/mL ascorbic acid 2-phosphate (Sigma–Aldrich, St. Louis, MO, USA), 40 μg/mL l-proline (Sigma–Aldrich), 1 μM dexamethasone (Sigma–Aldrich), 10% ITS^+^ Pre-mix Tissue Culture Supplement (Becton Dickinson), and 10 μg/mL transforming growth factor beta 1 (TGF-β1; Peprotech, Rocky Hill, NJ, USA). To make a pellet, aliquots of hDPSCs (5 × 10^5^ cells) were centrifuged at 500× *g* for 5 min in 15-mL conical tubes. Pellets were incubated in 5% humidified CO_2_ at 37 °C. The medium was changed every 2 to 3 days, and pellets were harvested at 3, 5, 7, 10, and 14 days of culture.

### 4.5. Cell Viability Assay

Cell viability was determined using the EZ-Cytox kit (Daeil Lab Service, Seoul, Korea), based on the water-soluble tetrazolium salt (WST) method. The hDPSCs were seeded in 96-well plates at a density of 1 × 10^4^ cells per well with various concentrations of *E. faecium* L-15 extract (0, 10, 25, 50, 100, and 200 μg/mL) for 72 h. Then, WST solution was added to each well. The mixture was incubated for 30 min at 37 °C. The absorbance of each well was measured at 450 nm with the Emax Plus Microplate reader (Molecular Devices, Sunnyvale, CA, USA).

### 4.6. Real-Time Quantitative Polymerase Chain Reaction (PCR)

Three independent replicates were conducted with two different batches of cells (A and B) at passage 6 and 7 (A passage 6, A passage 7, and B passage 6). Total RNA was extracted from pellets using the PureLink^TM^ RNA Mini kit (Life Technologies, Camarillo, CA, USA). The synthesis of cDNA was performed using M-MLV reverse transcriptase (Cosmogenetech, Seoul, Korea) according to the manufacturer’s instructions. Real-time PCR was performed using SYBR Pre-mix Ex Taq^TM^ ІІ (Takara, Tokyo, Japan) and the 7500 Real-Time PCR System (Applied Biosystems, Carlsbad, CA, USA). The primers used are listed in Appendix A. The PCR reaction was performed for 30 s at 95 °C, followed by 40 amplification cycles of 5 s at 95 °C and 34 s at 60 °C. The comparative C_T_ method was used to measure the level of expression. Glyceraldehyde 3-phophate dehydrogenase (*GAPDH*) was used as a housekeeping gene for normalization.

### 4.7. Quantitative Analysis of Glycosaminoglycan (GAG)

Four pellets from each group were used for GAG quantification. The amount of sulfated GAG was quantified using the Blyscan^TM^ glycosaminoglycan assay (Biocolor, Carrickfergus, UK) according to the manufacturer’s instructions. Briefly, pellets were digested for 12 h in papain extraction reagent at 65 °C. The lysate was mixed with Blyscan dye reagent and then with dissociation reagent. The absorbance at 525 nm was measured using the Emax Plus microplate reader (Molecular Devices). The relative cell number was determined by quantifying the total DNA using a Quant-iT^TM^ PicoGreen^TM^ dsDNA assay kit (Thermo Scientific, Waltham, MA, USA) according to the manufacturer’s instructions. GAG content was normalized by DNA content.

### 4.8. Quantification of Proteoglycan Content

The hDPSCs were differentiated into chondrocytes using chondrogenic differentiation medium and *E. faecium* L-15 extract for two weeks. Pellets were fixed with 4% paraformaldehyde for 30 min. Pellets were washed with phosphate-buffered saline (PBS) and incubated with alcian blue staining solution (Merck, Darmstadt, Germany) in the dark for 1 h at room temperature. Pellets were rinsed three times with ddH_2_O to neutralize the acidity. For quantitative analyses, alcian blue-stained cells were dissolved in 6 M guanidine hydrochloride (Sigma–Aldrich) for 6 h. The absorbance of the solubilized solution was measured at 650 nm.

### 4.9. Statistical Analysis

Results are presented as mean ± SD. Data were analyzed using one-way analysis of variance (ANOVA) followed by Dunnett’s test and two-way ANOVA with GraphPad Prism V5.0 software (GraphPad Software, La Jolla, CA, USA). *p* < 0.05 was defined as statistical significance.

## 5. Conclusions

We cultured the hDPSCs using chondrogenic differentiation medium in combination with L-15 extract and observed the effect of L-15 extract on chondrogenic differentiation. The expression of SOX9, COL2A1, ACAN, COL10A1, RUNX2, and MMP13, markers of chondrogenic differentiation, was analyzed and the amount of GAG or proteoglycan, an extracellular matrix of chondrocytes, was enumerated. The LET group showed that the markers of chondrogenic differentiation were over-expressed compared to the control group. In particular, these markers were over-expressed in the early stage. The amounts of GAG and proteoglycan were also found to be highly expressed in the LET group at all dates. This study suggests that L-15 extract promotes chondrogenic differentiation.

## Figures and Tables

**Figure 1 ijms-20-00624-f001:**
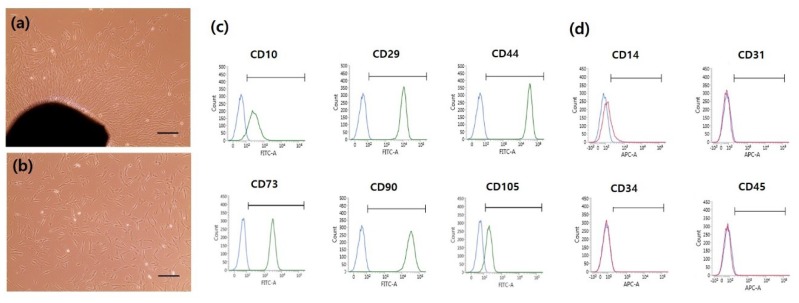
(**a**) The morphology of primary supernumerary tooth-derived human dental pulp stem cells (hDPSCs). (**b**) In vitro cultured hDPSCs at passage 3. The scale bar is 100 μm. (**c**) Characterization of hDPSCs at passage 4 by fluorescence-activated cell sorting (FACS) analysis. Mesenchymal stem cell markers (92.48% CD10; 100% CD29; 100% CD44; 100% CD73; 100% CD90; 88.13% CD105) were highly expressed in hDPSCs compared to (**d**) only a small degree of hematopoietic and endothelial marker expression (20.11% CD14; 0.53% CD31; 1.24% CD34; 0.82% CD45).

**Figure 2 ijms-20-00624-f002:**
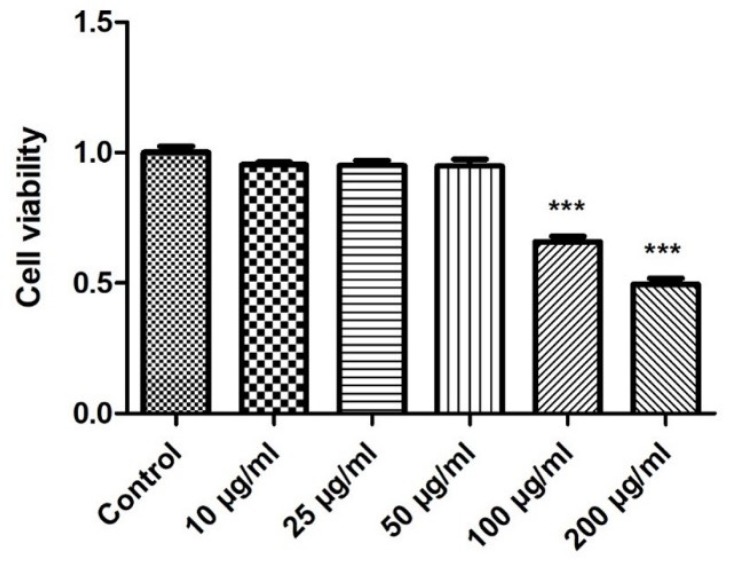
Water-soluble tetrazolium salt (WST) assays were used to detect hDPSC viability on exposure to L-15 extract (*n* = 3). Error bars represent mean ± S.D. *** *p* < 0.01, one-way ANOVA followed by Dunnett’s post hoc test was used.

**Figure 3 ijms-20-00624-f003:**
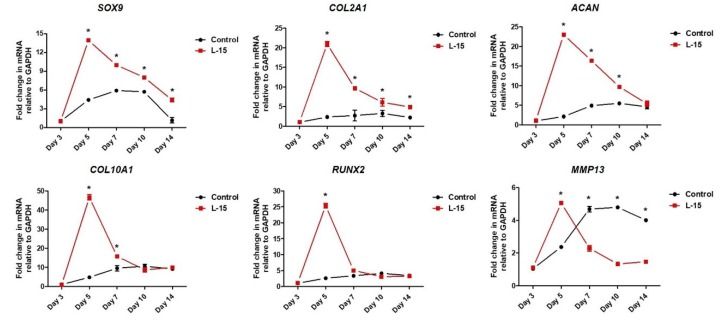
Total RNA was prepared from culture days 3, 5, 7, 10, and 14. The expression of early-stage chondrogenic marker genes (*SOX9*, *COL2A1*, and *ACAN*) and late-stage chondrogenic marker genes (*COL10A1*, *RUNX2*, and *MMP13*) was analyzed by qRT-PCR. *GAPDH* was used for normalization (*n* = 3). Error bars represent mean ± S.D. * *p* < 0.05, two-way ANOVA was used.

**Figure 4 ijms-20-00624-f004:**
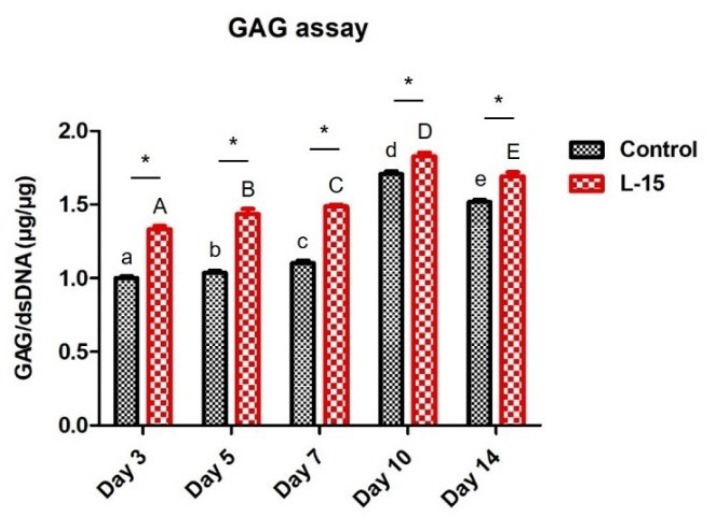
Total glycosaminoglycan (GAG) determinations were performed using the Blyscan^TM^ glycosaminoglycan assay (Biocolor, Carrickfergus, UK) in human DPSCs cultured in the presence or absence of 50 μg/mL L-15 extract (*n* = 3). The amounts of GAG were normalized by the amount of DNA contained in each sample. Error bars represent mean ± S.D. A, B, C, D, and E values of different superscripts indicate significant difference between LET groups and a, b, c, d, and e values of different superscripts indicate significant difference between control groups (* *p* < 0.05); two-way ANOVA was used.

**Figure 5 ijms-20-00624-f005:**
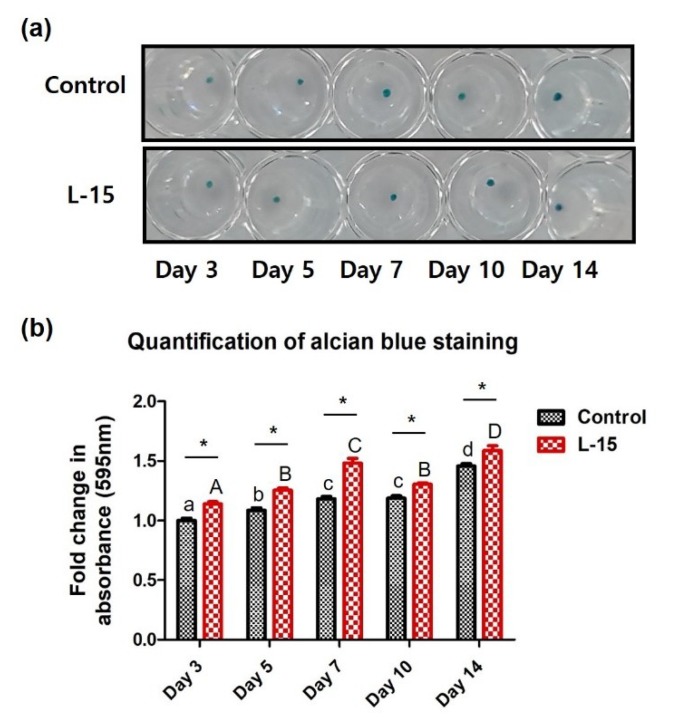
(**a**) Alcian blue staining of the control and L-15 extract-treated groups. (**b**) Quantitative measurements of alcian blue staining (*n* = 3). Error bars represent mean ± S.D. A, B, C, and D values of different superscripts indicate significant difference between LET groups and a, b, c, and d values of different superscripts indicate significant difference between control groups (* *p* < 0.05); two-way ANOVA was used.

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
