# Peer review of "Enterococcus faecium L-15 Cell-Free Extract Improves the Chondrogenic Differentiation of Human Dental Pulp Stem Cells"

_ijms, 2019, doi:10.3390/ijms20030624_

Reviewer 1 Report

In ‘Entercococcus faecium L-15 cell-free extract improves the chondrogenic differentiation of human dental pulp stem cells’, Kim et al. found that treatment of dental pulp stem cells with the L-15 extract promoted expression of chondrogenic markers at early time points in culture and increased extracellular matrix production. The novelty of the study is the use of the L-15 extract for chondrogenic differentiation. Overall, the study is interesting for the application of the lactic acid bacteria extracts; however, I have the following comments:

It would be interesting to know more about the hypothesized mechanism of action of the L-15 extract. Further, can it promote chondrogenic differentiation on its own, or are other medium supplements required?

The main evidence supporting the improvement in chondrogenic differentiation seems to be the qPCR data in Figure 3. All graphs show a substantial peak in the extract-treated condition at day 5, and I am wondering how reproducible this is. Were independent replicates measured (i.e., from different donors or by starting experiments on different days with different starting batches of cells)?

Because human primary cells were used, were the cells from the different donors cultured separately or were they pooled together? Is there an effect of donor on, for example, expression of MSC markers?

How was the cell viability assay performed? Were the cells grown in 2D culture for this experiment? With what starting density?

In Figure 4, are the error bars shown?

For the GAG and alcian blue assays, was an effect of time seen? That is, was a statistical analysis performed? At a first glance, it seems that there are little differences between days 3-7 in the GAG assay, for example. Is this really a steady increase (p. 4, line 111)?

p. 6, line 159: Are GAG and hyaluronic acid proteoglycans?

For the alcian blue staining, what is this detecting specifically?

Author Response

Response to reviewers' comments is attached as the file below.  

Reviewer 2 Report

The manuscript by Kim H et al. reported an effect of L15 extract on the chondrocyte differentiation from dental pulp stem cells (DPSC). The author tested safe dose of L15 extract on the DPSC survival and then examined the effect of L15 extract on the chondrocyte differentiation. By GAG assay and Alcan blue staining, they found that L15 extract positively regulated the chondrocyte differentiation of DPSC.

 Although most of the experiments were well-performed, and the manuscript is organized, there are several concerns before the publication in the International Journal of Molecular Sciences.

Specific comments,

1.    The authors showed the effect of L15 extract on the chondrocyte differentiation, and the results were statistically significant, however, the effects were seemingly subtle as shown in Figure 4 and 5b. For the chondrocyte differentiation, chondrocyte differentiation medium was used, so if L15 extract alone was treated to DPSCs, do they spontaneously differentiate into chondrocytes? Have the authors tested it?

2.    In Figure 2, Student’s t-test was used, but the graph showed multiple comparisons. In this case, the authors should use other statistics such as Dunnett's test or Bonferroni correction.

3.    Have the authors examined the effect of L15 extract on other mesenchymal stem cells such as bone marrow-derived or adipose-derived?

4.    Is there any reason why tested all the genes were overexpressed on Day 5 in Figure 3? These genes were remarkably overexpressed at gene levels. However, the phenotypic changes were relatively small. The authors should explain it.

5.    No error bars in Figure 4, please add.

Author Response

Response to reviewers' comments is attached below as file.

Round  2

Reviewer 1 Report

In their revision of ‘Entercococcus faecium L-15 cell-free extract improves the chondrogenic differentiation of human dental pulp stem cells’, Kim et al. have addressed a number of mine and the other reviewer’s comments.

I still feel that, particularly in the abstract and conclusions, it needs to be more clearly written that the L-15 extract was used in combination with a standard chondrogenic medium to stimulate chondrogenic differentiation. Also the statement ‘Specifically, our study is the first to confirm the molecular role of E. faecium L-15 in chondrogenic differentiation.’ seems to suggest something is known about the mechanism of action, although this is not the case.

Further, I have looked again at the qPCR results and have additional questions about how the data were analyzed. It seems that the data were normalized to the reading on Day 3 for both treatment conditions (this seems to be fixed at 1 in all graphs), so how is it possible to make comparisons between the two treatments, as they might already be different on Day 3 and the normalization will not take this into account?

My previous questions about the alcian blue staining had two reasons. The first is whether or not the alcian blue and Blyscan GAG assays are measuring the same thing (to be more clear in the discussion of the results). The second is whether or not these assays can be affected by the presence of the L-15 extract (e.g., if this contains any polysaccharide which can be binding to the pellets and influencing the assays).

Author Response

Please open the file attached below, thanks

Reviewer 2 Report

The authors addressed all the concerns. 

Author Response

Please open the file attached below, thanks.

Round  3

Reviewer 1 Report

The authors have addressed my comments.  Thank you.

Author Response

Thank you for accepting our manuscript and comments before.